# 4-Methylumbelliferone Inhibits Cancer Stem Cell Activation and Overcomes Chemoresistance in Ovarian Cancer

**DOI:** 10.3390/cancers11081187

**Published:** 2019-08-15

**Authors:** Noor A. Lokman, Zoe K. Price, Emily K. Hawkins, Anne M. Macpherson, Martin K. Oehler, Carmela Ricciardelli

**Affiliations:** 1Discipline of Obstetrics and Gynaecology, Adelaide Medical School, Robinson Research Institute, University of Adelaide, Adelaide, SA 5000, Australia; 2Department of Gynaecological Oncology, Royal Adelaide Hospital, Adelaide, SA 5005, Australia

**Keywords:** ovarian cancer, hyaluronan, chemotherapy resistance, cancer stem cells, 4-methylumbelliferone

## Abstract

We have recently shown that the extracellular matrix molecule hyaluronan (HA) plays a role in the development of ovarian cancer chemoresistance. This present study determined if HA production is increased in chemotherapy-resistant ovarian cancers and if the HA inhibitor 4-methylubelliferone (4-MU) can overcome chemoresistance to the chemotherapeutic drug carboplatin (CBP) and inhibit spheroid formation and the expression of cancer stem cell (CSC) markers. We additionally assessed whether 4-MU could inhibit in vivo invasion of chemoresistant primary ovarian cancer cells in the chicken embryo chorioallantoic membrane (CAM) assay. The expression of the HA synthases *HAS2* and *HAS3* was significantly increased in chemoresistant compared to chemosensitive primary ovarian cancer cells isolated from patient ascites. 4-MU significantly inhibited HA production, cell survival, and spheroid formation of chemoresistant serous ovarian cancer cells. In combination with CBP, 4-MU treatment significantly decreased ovarian cancer cell survival and increased apoptosis of chemoresistant primary cells compared to CBP alone. 4-MU significantly reduced spheroid formation, expression of CSC markers *ALDH1A1* and *ABCG2* in primary cell spheroid cultures, and ALDH1 immunostaining in patient-derived tissue explant assays following treatment with CBP. Furthermore, 4-MU was very effective at inhibiting in vivo invasion of chemoresistant primary cells in CAM assays. Inhibition of HA is therefore a promising new strategy to overcome chemoresistance and to improve ovarian cancer survival.

## 1. Introduction

Ovarian cancer is the most lethal gynaecological cancer and the fifth most common cause of cancer-related death among Western women [1]. Current clinical practice for advanced ovarian cancer consists of de-bulking surgery followed by combined platinum and taxane-based chemotherapy. High grade serous ovarian cancers make up nearly 70% of ovarian cancers and are characterized by high initial chemosensitivity. However, 75% of patients relapse after treatment and subsequently develop resistance to platinum-based drugs [2]. Chemotherapy resistance is the major clinical challenge in ovarian cancer treatment and strategies that can overcome chemoresistance will greatly improve patient survival.

Chemoresistance is multifactorial involving both tumour and drug-related factors, but recent data also suggest a critical role of the tumour microenvironment [3]. Our recent studies focussing on the tumour microenvironment linked chemoresistance to the production of the extracellular matrix (ECM) molecule hyaluronan (HA), which is known to play an important role in cancer progression and metastasis [4]. HA is a large polysaccharide which is assembled into pericellular and ECM matrices in many tissues [5]. HA levels are regulated by three HA synthases (HAS1–HAS3) which synthesize HA from glycolytic metabolites UDP-D-glucuronic acid (GLcUA) and UDP-N-acetylglucosamine (GlcAc) [6]. HA levels in mammalians are also controlled by three HA-degrading hyaluronidases (HYAL1–3) which produce low molecular weight HA fragments in the ECM [7]. HA plays a role in various cell functions such as adhesion, motility, and differentiation and has been implicated in playing a key role in cancer metastasis [5,8] with HA levels correlating with the degree of invasiveness and metastatic potential of ovarian cancers [9].

HA treatment can reduce the ability of chemotherapy drugs to cause cancer cell death in various cancer cell lines [10,11,12]. We recently investigated the ability of HA to block the growth inhibitory effects of the cytotoxic drug carboplatin (CBP) in a range of ovarian cancer cells [13]. We found that adding HA to CBP-treated cells only increased survival of ovarian cancer cell lines expressing the HA receptor CD44 [13]. The increased cell survival observed with concurrent CBP and HA treatment was reversed by blocking CD44-HA interactions by either the addition of HA oligomers (chains of 6–10 saccharides), or a neutralising CD44 antibody [13].

A potent inhibitor of HA synthesis is 4-methylumbelliferone (4-MU). 4-MU reduces the cellular content of UDP-GlcUA and also inhibits synthesis of HA synthase enzymes [14]. 4-MU (also known as hymecromone) is a modified coumarin of plant origin which has been used in patients in Asia and Europe for the treatment of hepatobiliary disease because of its choleretic and biliary antispasmodic activity [15]. 4-MU is effective at blocking growth and metastasis of several different cancers including pancreatic, prostate, breast and liver cancers in animal studies [16,17,18,19,20]. Limited studies to date have investigated the effects of 4-MU on ovarian cancer cells.

This study assessed if HA production is increased in patients developing chemoresistant disease and in chemoresistant primary serous ovarian cancer cells compared to chemosensitive cells. We determined whether the HA inhibitor 4-MU could increase the cytotoxic effect of CBP. We also evaluated whether 4-MU could inhibit spheroid formation and expression of cancer stem cell (CSC) markers in spheroids and a patient-derived explant tissue assay [21], as well as in vivo invasion of chemoresistant primary ovarian cancer cells using the chicken embryo chorioallantoic membrane (CAM) assay [22].

## 2. Results

### 2.1. HA Serum Levels Are Elevated in Patients with Chemoresistant Disease

Serum HA levels are significantly elevated in serous ovarian cancer patients that develop chemoresistant disease (median 123.2, range 58.6–252 ng/mL) compared to levels at diagnosis (median 21.3, range 10.5–111.7 ng/mL) (Figure 1a, *n* = 9, *p* = 0.0039, Wilcoxon pair test). In contrast, serum HA levels were not significantly elevated in patients who relapsed but continued to respond to chemotherapy treatment (Figure 1b, *n* = 7, *p* = 0.219, Wilcoxon pair test). HA staining in matching tissues from two patients at diagnosis confirms increased production of HA in cancer cells and the peritumoral stroma following relapse with chemotherapy-resistant disease (Appendix A).

### 2.2. HA Production Is Increased in Serous Ovarian Cancer Cells Following Development of Chemotherapy Resistance

We examined expression of HA synthases (*HAS1*, *HAS2*, and *HAS3*) and hyaluronidases (*HYAL1, HYAL2*) in primary serous ovarian cancer cells isolated from patient ascites and CBP-resistant OV-90 cells. We found that *HAS2* (Figure 2b) and *HAS3* (Figure 2c) but not *HAS1* (Figure 2a) expression is significantly increased in primary serous ovarian cancer cells isolated from the ascites of patients with chemoresistant disease compared to patients with chemosensitive disease. *HAS2* and *HAS3* expression was also significantly increased in CBP-resistant OV-90 CBPR cells compared to parental cells (Figure 2b,c). *HAS1* was not detected in any ovarian cancer cell lines examined. *HYAL1* and *HYAL2* expression was not different between the chemosensitive and chemoresistant primary ovarian cancer cells nor between CBP-resistant OV-90 cells compared to parental cells (Figure 2d,e). We also confirmed by HA ELISA that chemoresistant primary serous ovarian cancer cells had significantly higher levels of HA in the conditioned media compared to chemosensitive cells (Figure 2f). HA levels were also significantly increased in conditioned media from OV-90 CBPR cells compared to parental OV-90 cells (Figure 2f).

### 2.3. 4-MU Treatment Inhibits Survival of Chemoresistant Ovarian Cancer Cells

We investigated whether 4-MU could inhibit the survival of ovarian cancer cells (as measured by cell viability with the 3-(4,5-dimethylthiazol-2-yl)-2,5-diphenyl tetrazolium bromide, MTT assay) including primary chemosensitive and chemoresistant serous cancer cells derived from patient ascites and established ovarian cancer cell lines (OV-90, OV-90 CBPR, SKOV3) with varying sensitivity to CBP [13]. Established cell lines with CBP (half maximal inhibitory concentration, IC_50_ ≥180 µM (SKOV3, OV-90 CBPR) were classified as chemoresistant and primary cells from patient ascites were classified as chemosensitive or chemoresistant based on clinical response to chemotherapy. We initially tested a range of 4-MU concentrations (0–1 mM) and found that 1 mM, but not lower concentrations of 4-MU (0.1 mM, 0.5 mM), could significantly inhibit the survival of OV-90, SKOV3 cells, chemosensitive (P9), and chemoresistant (P13) primary serous ovarian cancer cells (Appendix A). We used 1 mM 4-MU for all subsequent experiments. 4-MU (1 mM) significantly reduced the cell survival of both chemosensitive (black bars, range 43–69% of control Figure 3a) and chemoresistant (grey bars, range 58–82% of control, Figure 3a) ovarian cancer cells. 4-MU was very effective at inhibiting HA production in both chemosensitive (OV-90, P9) and chemoresistant ovarian cancer cells (SKOV3, OV-90 CBPR, P13, Figure 3b). However, exogenous HA (222 kDa and 1110 kDa, 10 µg/mL) treatment could not reverse the effects of 1 mM 4-MU in the MTT assays (Appendix A).

We assessed whether 4-MU could increase the cytotoxic effect of CBP in both chemosensitive and chemoresistant ovarian cancer cells. In cell survival assays, OV-90 cells and chemosensitive primary ovarian cancer cells exhibited a similar response to CBP (100 µM) or 4-MU (1 mM). Combined 4-MU and CBP treatment did not further decrease ovarian cancer cell survival (Figure 3c,d). However, combined CBP (100 µM) and 4-MU treatment (1 mM) significantly decreased survival of SKOV-3 cells and chemoresistant primary serous ovarian cancer cells (*n* = 8) compared to CBP alone (Figure 3e,f). Effects of 4-MU (1 mM) and/or CBP (100 µM) treatment on apoptosis were measured by using a caspase 3/7 cleavage assay in primary serous ovarian cancer cells. We showed that 4-MU alone and combined CBP and 4-MU treatments increased the level of apoptosis in chemosensitive primary cells compared to control treatment (Figure 3g). In chemoresistant primary cells, combined CBP and 4-MU treatment increased the level of apoptosis compared to cells treated with control media or CBP alone (Figure 3h).

### 2.4. 4-MU Inhibits Spheroid Formation and Stem Cell Marker Expression

4-MU treatment significantly reduced the area of spheroids formed by OV-90 (Figure 4a), and chemoresistant primary ovarian cancer cells (Figure 4b). OV-90 cells exhibited a similar response to CBP (100 µM) or 4-MU (1 mM) treatment alone and combined 4-MU and CBP treatment did not further decrease OV-90 spheroid size (Figure 4a). CBP treatment had no effect on the spheroid area of chemoresistant primary serous ovarian cancer cells. However, in combination with 4-MU (1 mM), the spheroid area of chemoresistant cells was significantly reduced compared to CBP alone (Figure 4b).

We showed by qRT-PCR that 4-MU treatment reduced expression of both *HAS2* (Figure 5a) and *HAS3* (Figure 5b) in OV-90 spheroids and spheroids formed by primary chemoresistant cells. Expression of CSC marker *PROM1* was significantly reduced in OV-90 spheroids (Figure 5c). *PROM1* was not expressed by the primary chemoresistant cell spheroids. *ALDH1A1* (Figure 5d) and *ABCG2* expression (Figure 5e) was significantly reduced in spheroid cultures from chemoresistant primary cells following 4-MU treatment. CD44 expression in the spheroids was not affected by 4-MU treatment (Figure 5f).

### 2.5. ALDH1 Protein Expression is Increased Following Chemotherapy Treatment and Reduced by 4-MU Treatment in Patient-Derived Explant Assays

We assessed ALDH1 immunostaining in serous ovarian cancer tissues from patient derived tissue explant assays (*n* = 5). ALDH1 immunostaining (percentage positive area, % POS area) measured by video image analysis was significantly increased following 48 h of treatment with CBP (100 µM) compared to control tissues (Figure 6a). However, the increase in ALDH1 % POS area by CBP treatment was not observed in tissues treated with both 4-MU (1 mM) and CBP (100 µM) (Figure 6a). Representative examples of the ALDH1 immunostaining in ex vivo explant tissues treated with control media, CBP, 4-MU, or combined 4-MU and CBP are shown in Figure 6b–e, respectively. We also showed that ALDH1 protein expression is increased in serous ovarian cancer tissues following chemotherapy and at relapse compared to the levels in chemonaïve tissues (Figure 6f). Increased ALDH1 immunostaining is evident in the images of Figure 6h,j that are examples of tissues from the same patient after chemotherapy treatment and recurrence, respectively, compared to matching chemonaïve tissues obtained at time of diagnosis (Figure 6g,i).

### 2.6. 4-MU Treatment Reduces In Vivo Invasion of Chemoresistant Ovarian Cancer Cells

We evaluated the effects of 4-MU treatment on chemoresistant primary serous ovarian cancer cell invasion using the in vivo chick chorioallantoic membrane (CAM) assay. CD44 immunohistochemistry was used to assess ovarian cancer cell invasion from the ectoderm into the mesoderm layer of the CAM. Representative images of the cell invasion following treatment with control media, 4-MU (1 mM), CBP (100 µM) and CBP + 4-MU are shown in Figure 7a–d. Quantitative analysis showed that 4-MU (1 mM) alone or combined with CBP (100 µM) treatment significantly decreased invasion of chemoresistant primary cells compared to control treatment (Figure 7e).

## 3. Discussion

The major limitation to the successful treatment of serous ovarian cancer is the emergence of chemotherapy resistance. The development of more effective therapies to overcome chemotherapy resistance and improve survival is urgently required. In this study we found that the expression of the HA synthases *HAS2* and *HAS3* was significantly increased in chemoresistant compared to chemosensitive primary ovarian cancer cells isolated from patient ascites. 4-MU significantly inhibited HA production, cell survival, and spheroid formation of chemoresistant serous ovarian cancer cells. Furthermore, 4-MU was effective at inhibiting expression of CSC markers (*ALDH1A1* and *ABCG2*) and in vivo invasion of chemoresistant serous ovarian cells. Our findings indicate that 4-MU treatment is a promising strategy to inhibit HA production and CSC activation and improve survival of patients with serous ovarian cancer.

In a previous report we showed that serum HA levels are increased in patients following chemotherapy and patients that develop recurrent disease [13]. In this current study using matched serum samples at diagnosis and relapse, we demonstrated that HA serum levels are significantly increased up to 5-fold in serous ovarian cancer patients that relapsed with chemoresistant disease but not in patients that relapsed but continue to respond to chemotherapy treatment. We showed that HA was increased in both the peritumoral stroma and serous ovarian cancer cells in matching tissues from patients at diagnosis and following relapse with chemoresistant disease. We additionally showed that expression of *HAS2* and *HAS3* but not *HAS1* or hyaluronidases *HYAL1* and *HYAL2* were significantly increased in chemoresistant compared to chemosensitive primary serous ovarian cancer cells. *HAS2* and *HAS3* were similarly increased in OV-90 CBP cells that acquired CBP resistance compared to parental OV-90 cells. Recent studies have also shown that leukemic cell lines with increased HA production resist chemotherapy [23,24]. Together, these findings support the concept that HA plays an important role in the development of acquired chemotherapy resistance.

A potential mechanism whereby HA mediates chemoresistance is via expression of ABC transporter membrane proteins which decrease levels of chemotherapy drugs within cells [25]. Several studies have demonstrated that HA-CD44 interactions mediate chemotherapy resistance by regulating the expression and activity of ABC transporters which function as efflux pumps and interfere with the intracellular accumulation and retention of chemotherapy drugs [11,26]. We have previously demonstrated that HA can regulate the expression of multiple ABC transporters including *ABCB3*, *ABCC1*, *ABCC2*, and *ABCC3* and carboplatin treatment increased *ABCC2* and HA in OVCAR-5 cells [13]. In breast cancer, overexpression of HAS2 was reported to stimulate *ABCB1* expression through the PI3K pathway, increasing resistance to doxorubicin [27]. It was found that 500-kDa HA stimulated *ABCB1* expression via CD44 in breast cancer (MCF-7 cells), inducing resistance to doxorubicin, paclitaxel, and etoposide [11,28]. HA (molecular weight not specified) also promoted expression of *ABCC2* in non-small cell lung cancer cells [29].

Our findings that 4-MU can inhibit HA production and cell survival and promote apoptosis of serous ovarian cancer cells agrees with other cancer studies in the literature [16,30,31,32,33,34,35,36,37]. We have shown in this study that 4-MU is effective at inhibiting HA production and survival of both chemosensitive and chemoresistant primary serous ovarian cancer cells. However, in chemoresistant serous ovarian cancer cells, combined 4-MU and CBP treatment was more effective at inhibiting cell survival compared to CBP alone. These findings also concur with our recent studies using ex vivo tissue explant assays demonstrating that combined 4-MU and CBP treatment significantly increased apoptosis and reduced proliferation in ovarian cancer tissues from patients with chemoresistant disease [21].

To date there have been limited studies that have investigated effects of 4-MU on ovarian cancer cells. Previous studies have shown 4-MU treatment inhibits HA production and spheroid formation by HAC-2 ovarian clear cell carcinoma cells [38] and *HAS3* expression by SKOV3 ovarian cancer cells [14]. Tamura et al. recently reported that 4-MU inhibited cell proliferation of HRA ovarian cancer cells in vitro [18]. However, as HRA ovarian cancer cells were found to express very low levels of *HAS2* and *CD44*, the in vitro effects of 4-MU in these cells may be via a mechanism other than HA-CD44 signaling [18]. Our findings that the addition of exogenous HA could not reverse the effects of 1 mM 4-MU in the MTT assays supports this finding that 4-MU may also have anti-tumour activity that is not dependant on HA. The study by Lompardia et al. 2013 [23] found that co-treatment with 500 µM 4-MU and 30-fold higher levels of HA (300 µg/mL) partially reverted the effects of 4-MU in leukemic K5652 and Kv563 cells. Whereas treatment with lower concentration of 4-MU (100 µM) and HA 300 µg/mL completely restored baseline conditions in the cell lines. The authors concluded that effects of low concentrations of 4-MU could be restored by exogenous HA and but higher doses of 4-MU may trigger anti-proliferative signals independent of HA. In another study Arai et al. 2011 also found that exogenous HA (200 µg/mL) could not neutralize effects of 4-MU on osteosarcoma cells including formation of cell matrix, or cell proliferation [31]. They concluded from this finding that HA may have different biological activity if presented to cells as free exogenous HA or as endogenous cell-associated HA. Keller et al. found that 4-MU reduced both versican and fibronectin in trabecular meshwork cells of the eye [39] but the effects could not be reversed by the addition of exogenous high (1500 kDa) or low (40 kDa) molecular weight HA (500 µg/mL) to the culture medium. They suggested that since exogenous HA could not reverse effect of 4-MU, only de novo synthesized HA altered versican and fibronectin levels. It is likely that 4-MU may affect the synthesis and organization of other ECM components to mediate its anti-proliferative effects on ovarian cancer cells.

The stemness and expansion of CSCs are thought to be highly influenced by changes in the microenvironment and recent studies have highlighted a key role for HA in regulating CSC populations [40,41]. Excessive HA production allows breast cancer cells to revert to a stem cell state via the up-regulation of genes involved in regulating epithelial-mesenchymal transition [41]. HA has also been shown to promote the formation of CSC populations in breast cancer [41] and glioblastoma cell lines [42]. Additionally, HA activates genes associated with stemness in embryogenesis and interacts with CSCs to enhance stemness and therapy resistance [43]. Shiina et al. showed that molecular weight of HA was important in promoting and maintaining stemness of CSCs in the head and neck cancer cell line HSC-3 [44]. It was found that 200 kDa HA significantly promoted expression of cancer stem cell genes, spheroid formation, and cisplatin resistance in ALDH^high^ CD44v3^high^ HSC-3 cells compared to 5, 20, and 700 kDa HA [44]. Previous studies by Okuda et al. 2012 showed that 4-MU treatment reduced HA matrix produced from CSCs isolated from MDA-MB231 cells overexpressing *HAS2* [40]. Studies by Hiraga et al. 2013 and Chanmee et al. 2014 also found that 3D growth of breast cancer cells was inhibited by 4-MU in a dose-dependent manner [41,45]. In this study we found that 4-MU inhibited spheroid formation of both chemosensitive and chemoresistant serous ovarian cancer cells. Furthermore, we showed that 4-MU inhibited expression of *HAS2, HAS3* and CSC markers (*ALDHA1* and *ABCG2*) in a 3D-spheroid culture of chemoresistant primary serous ovarian cancer cells. However, the expression of other CSC markers identified in serous ovarian cancer cells [46] including *PROM1* (CD133) and *CD44* were not affected by 4-MU treatment in spheroids formed by the primary serous ovarian cancer cells. A recent study has shown that 4-MU treatment significantly reduced expression of CD44, CD133, CD90 and EpCAM in hepatic carcinoma cells in vitro [47].

Several ovarian cancer studies reported CSC marker expression to be increased following chemotherapy [48,49,50,51,52,53]. We found that ALDH1 protein levels were significantly increased in cultured explant tissues by CBP alone and this could be prevented by treatment with 4-MU. We also observed increased ALDH1 positivity in serous ovarian cancer tissues obtained from patients that had received neoadjuvant chemotherapy or had relapsed compared to chemonaïve tissues from patients who had not received chemotherapy. In agreement with these findings, previous studies also found that ALDH1 expression and activity was significantly higher in taxane- and platinum-resistant ovarian cancer cells [50,51]. ALDH1-positive ovarian cancer cells were also enriched in residual A2780 tumour xenografts after platinum therapy [52] and in patients that had received neo-adjuvant chemotherapy treatment [53].

We demonstrated that 4-MU treatment inhibits the in vivo invasion of chemoresistant primary serous ovarian cancer cells using the CAM assay. To the best of our knowledge this is the first study to assess cell invasion of primary ovarian cancer cells in vivo using the CAM model. The inhibitory effects on serous ovarian cancer cell invasion concur with previous in vitro [14,31,32,35,36,37,40,54,55] and in vivo [17,30,31,32,35,36,40,55,56,57] studies investigating the anti-tumour effect of 4-MU. The only ovarian cancer study to date investigating effects of 4-MU on motility and invasion found no effect of 4-MU, either on HRA ovarian cancer cell migration nor invasion in vitro, but reported that 4-MU treatment inhibited HRA tumour growth and metastasis in a rat model in vivo [18]. It is likely that the tumour inhibitory effects of 4-MU in vivo may be mediated by inhibiting HA in the tumour microenvironment as HRA ovarian cancer cells expressed low levels of *HAS2* and *HAS3* [18].

## 4. Materials and Methods

### 4.1. Cell Line Culture

The human ovarian cancer cell lines SKOV-3 and OV-90 were purchased from American Type Culture Collection (ATCC, Manassas, VA, USA) and authenticated by a short tandem repeat (STR) DNA profile in 2016. All cell lines were grown in RPMI 1640 media (cat no. R8758, Sigma Aldrich, St. Louis, MO, USA) and cultured with 10% foetal bovine serum (Sigma Aldrich) and maintained at 37 °C in an environment of 5% CO_2_. OV-90 cells were made resistant to CBP following treatment with 6–8 cycles of CBP (50 µM, Accord Healthcare Pty Ltd, Melbourne, VIC, Australia) [58]. The OV-90 CBPR cells exhibited an IC_50_ to CBP that was two-fold higher than that for the parental cells (data not shown).

### 4.2. Primary Cell Isolation

Primary ovarian cancer cells were derived from ascites collected from advanced stage ovarian cancer patients prior to chemotherapy treatment (*n* = 9) and following the development of chemoresistant recurrent disease (*n* = 11) with approval of the Royal Adelaide Hospital Human Ethics Committee as described previously [58]. All primary cells were grown in Advanced RPMI 1640 medium (cat no. 12633-020, Life Technologies, Mulgrave, VIC, Australia) supplemented with 4 mM L-glutamine, 10% FBS (Sigma Aldrich, St Louis, MO, USA), and antibiotics (100 U penicillin G, 100 µg/mL streptomycin sulfate, and 100 µg/mL amphotericin B, Sigma Aldrich). The epithelial nature of the primary ovarian cancer cells was confirmed by cytokeratin immunocytostaining [59]. The clinicopathological characteristics of the patients used to isolate the primary cells are shown in Appendix A. Patient response to adjuvant chemotherapy was recorded after six months of treatment. Patients were classified as chemosensitive if they exhibited a complete response and did not progress within 6 months after completing the chemotherapy treatment. Patients that relapsed were classified as chemoresistant when they longer responded to chemotherapy treatment.

### 4.3. HA Detection

An HA ELISA kit (cat no. DY3614, R&D Systems) was used to determine the concentration of HA in serum samples or conditioned media samples as per manufacturer’s instructions [13]. HA in tissues was detected as described previously [13]. HA staining was measured by video image analysis (VIA, VideoPro 32; Leading Edge P/L, Marion, SA, Australia) as described previously [60]. Colour images from contiguous fields for each tissue core were collected at a magnification of ×400. VIA measurements included the DAB stained area (i.e., positively stained area in pixel units) and the total tumour area examined (i.e., positively and negatively stained area in pixel units), for each field, were used to derive the % POS area. The clinicopathological characteristics of the relapsed patients used for HA serum measurements are shown in Appendix A.

### 4.4. Cell Survival Assays

Ovarian cancer cells were plated at 5000 cells/well in 96-well plates in respective growth media. After 24 h, cells were treated with control media, 4-MU (1 mM, cat no. M1508, Sigma Aldrich), CBP (100 µM, Accord Healthcare Pty Ltd) or 4-MU (1 mM) + CBP (100 µM) for 72 h. This concentration of 4-MU has been shown to inhibit HA production by 60 to 80% in primary ovarian cancer cells (data not shown). 4-MU was supplemented in the culture media every 24 h. Cell survival was calculated by MTT assay as per the manufacturer’s instructions (Sigma Aldrich) [13].

### 4.5. Apoptosis Assays

Primary ovarian cancer cells were plated at 10,000 cells/well in 96-well plates in growth media. After 24 h, cells were treated with control media, 4-MU (1 mM), CBP (100 µM), or 4-MU (1 mM) + CBP (100 µM) for 72 h. Cell apoptosis by caspase 3/7 cleavage was measured using Caspase Glo 3/7 assay (Promega, Madison, WI, USA) as per manufacturer’s instructions. Luminescence endpoint was measured using a Synergy H1 plate reader (BioTek Instrument Inc. Winooski, VT, USA).

### 4.6. Spheroid Assays

Primary ovarian cancer and OV-90 cells (2.5 × 10^5^ cells/well) were plated on poly-HEMA (30 mg/mL in 95% ethanol, cat no. P3932, Sigma Aldrich) coated 24-well culture plates in RPMI containing 10% FBS. Following 24 h cells were treated with control media or 4-MU (1 mM), CBP (100 µM), or 4-MU (1 mM) + CBP (100 µM). Spheroid formation was observed over four days. Spheroid formation was assessed in images taken using a light microscope EVOS^®^ FL Imaging System (Life Technologies) using a 4× objective. The area (µm^2^) of spheroids greater than 150 μm in length (*n* = 5 images) was determined for each of the treatment groups using Image J32 software, (Image J I.50i, National Institute Health, Bethesda, MD, USA).

### 4.7. Quantitative RT-PCR

Primary ovarian cancer cells were plated at 30000 cells/well in 96-well plates and cultured for 24 h. RNA was extracted and reverse-transcribed with the TaqMan^®^ Gene expression Cells-to-CT kit (Applied Biosystems, Waltham, MA USA), according to manufacturer’s instructions. For spheroid cultures, RNA was isolated using the mirVana miRNA Isolation Kit (AM1560, Thermofisher, Waltham, MA USA), as per the manufacturer’s instructions. RNA from spheroid culture was quantified using NanoDrop (Thermofisher), and 20 ng were added in complementary DNA cDNA synthesis reactions using TaqMan^®^ Gene expression Cells-to-CT kit. qRT-PCR reactions were performed on triplicate cDNA samples using TaqMan^®^ primer sets in Appendix A using Quantstudio 12 K flex real time PCR machine (Applied Biosystems) as described [58]. CT values were normalised to the house keeping gene β-actin and calibrator using the 2^−∆∆CT^ method.

### 4.8. ALDH1 Immunohistochemistry

Formalin fixed serous ovarian cancer tissue sections were obtained from ex vivo tissue explant assays (*n* = 5) treated for 48 h with control media, 4-MU (1 mM), CBP (100 µM) or a combination of 4-MU (1 mM) and carboplatin (100 µM) [21]. Tissue from chemotherapy-naïve serous ovarian cancer patients at surgery (*n* = 15), after receiving neoadjuvant chemotherapy (*n* = 20) and at relapse (*n* = 5), were collected with approval from the Royal Adelaide Hospital Human Ethics Committee after informed consent. The clinicopathological characteristics of the patients used to for ex-vivo tissue explant assays and patient cohort used for ALDH1 immunohistochemistry are shown in Appendix A respectively. All tissue sections (5 μm) underwent microwave antigen retrieval as described previously [58] and incubated overnight with ALDH1 mouse monoclonal antibody (1/400, clone 44/ALDH1, BD Biosciences) in blocking buffer (5% normal goat serum) at 4 °C. Visualization of immunoreactivity was achieved using biotinylated anti-mouse immunoglobulins, streptavidin-peroxidase conjugate and diaminobenzidine substrate as described previously [58]. Negative controls included tissues incubated with no primary antibody or with mouse immunoglobulins. Slides were digitally scanned using the NanoZoomer Digital Pathology System (Hamamatsu Photonics, Hamamatsu, SZK, Japan) and images were collected using NDP view imaging software (NDP scan software v2.2, Hamamatsu Photonics). ALDH1 immunostaining was measured by VIA as for HA described previously [60].

### 4.9. CAM In Vivo Invasion Assays

Fertilized white leghorn chicken eggs were obtained from Hi-Chick, South Australia, Australia. Eggs were incubated in a MultiQuip Incubator at 37 °C with 60% humidity. Ethics approval was obtained by the University of Adelaide Animal Ethics Committee (RAH protocol No: 140101 and R20181215). Primary serous ovarian cancer cells derived from patient’s ascites with chemoresistant disease (9 × 10^5^) were mixed with Matrigel (8.9 mg/mL, BD Biosciences) containing vehicle (PBS), 4-MU (1 mM), CBP (100 µM), or a combination of 4-MU (1 mM) and CBP (100 µM) and placed on top of the CAM of day 11 chick embryos [22]. Matrigel grafts with adjacent CAM were harvested from each embryo (*n* = 6–9/treatment group) after 3 days (day 14), fixed with 10% formalin for 24 h, processed and embedded in paraffin. Serial sections (6 µm) were stained with haematoxylin and eosin or immunostained with CD44 antibody (1/800, Clone Ab-4, 156-3C11, Thermo Scientific, Lab Vision Corporation, Fremont, CA, USA). Immunohistochemistry was performed as described previously using citrate buffer antigen retrieval [58]. Slides were digitally scanned using the NanoZoomer (Hamamatsu Photonics). The area of the CD44 positive cancer cells that invaded through the ectoderm into the mesoderm was measured using the NanoZoomer area tool. Data are expressed as CD44-positive areas (µm^2^/mm^2^ of mesoderm).

### 4.10. Statistical Analyses

All statistical analyses were performed using Graph Pad Prism 7 (version 7.02, La Jolla, CA, USA). The Wilcoxon pair test was used to assess differences in serum HA levels in matching samples at diagnosis and after development of recurrence. As data were not normally distributed, the Mann Whitney U test was used to assess differences between gene expression in chemosensitive and chemoresistant primary serous ovarian cells, and HA levels in conditioned media from primary chemosensitive and chemoresistant serous ovarian cancer cells. The independent *t* test or one-way ANOVA with Tukey’s multiple comparisons tests was used to assess differences between parental and CBP-resistant OV-90 cells and the treatment groups in the cell survival assays. The Friedman test with Dunn’s multiple comparison test was used to assess differences between treatment groups in the ex-vivo explant assay. The Kruskal–Wallis test with Dunn’s multiple comparison test was used to assess differences between the control and treatment groups for spheroid area, gene expression in spheroids, immunostaining for ALDH1, and CAM invasion. Statistical significance was accepted at *p* < 0.05.

## 5. Conclusions

In summary, our results show that 4-MU is effective at inhibiting the growth and invasion of chemoresistant serous ovarian cancer cells. 4-MU is able to inhibit several HA-mediated neoplastic properties such as proliferation and invasion by inhibiting HA production, inducing apoptosis, and inhibiting CSC activation in chemoresistant cells. The experience in both humans and animals to date suggests that 4-MU is safe and well tolerated [15,32,56]. Our findings suggest that 4-MU treatment may be effective to inhibit HA production and CSC activation which occurs following chemotherapy treatment. 4-MU treatment is therefore a promising strategy to improve survival of patients with chemoresistant ovarian cancer and warrants further investigation in pre-clinical models of ovarian cancer.

## Figures and Tables

**Figure 1 cancers-11-01187-f001:**
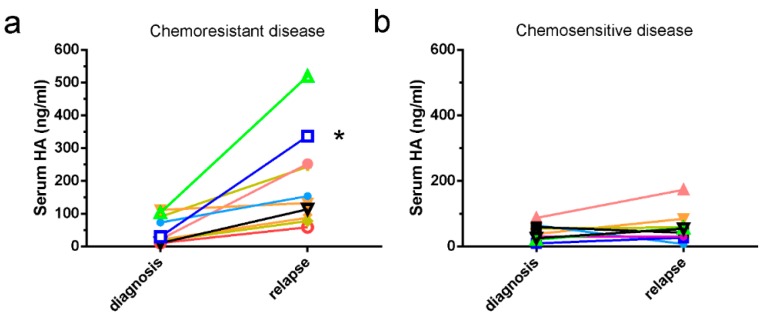
Serum hyaluronan (HA) is elevated in patients with chemoresistant disease. (**a**) HA serum levels (ng/mL) in serous ovarian cancer patients at initial diagnosis and following relapse with chemoresistant disease (*n* = 9). * significantly different from levels at diagnosis (*p* = 0.0039, Wilcoxon pair test). (**b**) HA serum levels (ng/mL) in serous ovarian cancer patients at initial diagnosis and following relapse with chemosensitive disease (*n* = 7, *p* = 0.219, Wilcoxon pair test).

**Figure 2 cancers-11-01187-f002:**
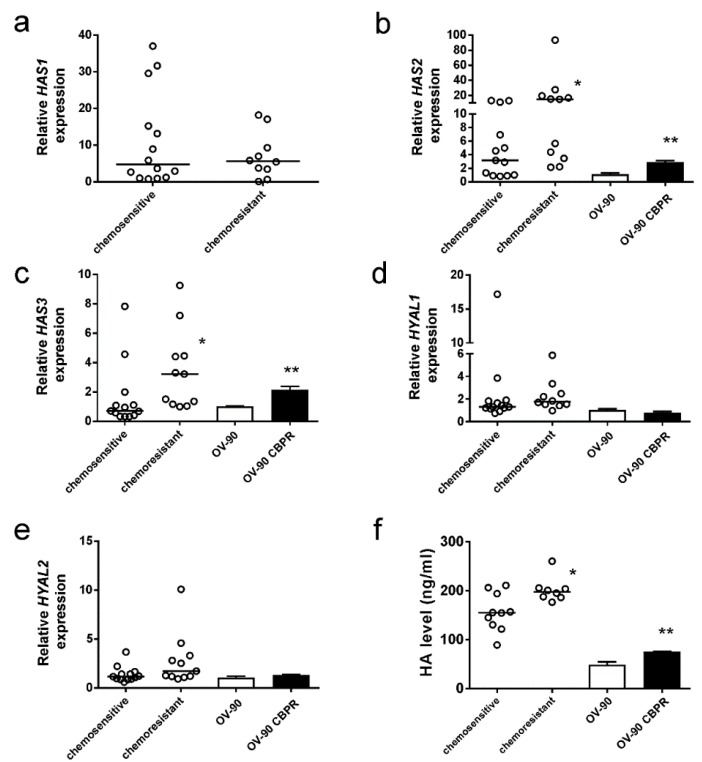
Hyaluonan (HA) synthase and hyaluronidase expression and HA production in chemosensitive and chemoresistant serous ovarian cancer cells. Expression in chemotherapy-resistant primary serous ovarian cancer cells compared to chemotherapy-sensitive cells and OV-90 cells made resistant to carboplatin (OV-90 CBPR). *HAS1* (**a**), *HAS2* (**b**) *HAS3* (**c**), *HYAL1* (**d**), and *HYAL2* (**e**) *, *HAS2* (*p* = 0.0218, Mann Whitney U test) and *HAS3* (*p* = 0.0107, Mann Whitney U test) but not *HAS1* expression (*p* = 0.879, Mann Whitney U test) was significantly increased in chemoresistant cells compared to chemosensitive cells. **, *HAS2* (*p* = 0.021, Student *t* test) and *HAS3* (*p* < 0.0001, Student *t* test) were significantly increased in OV-90 CBPR compared to parental cells. *HYAL1* and *HYAL2* expression was not significantly different between the chemosensitive and chemoresistant primary cancer cells nor the OV-90 cell lines. The bars for the primary cells specify the median values in each group and are expressed as the mean fold change from RNA samples (*n* = 6–9) from three independent experiments. Data for OV-90 cells are expressed as the mean fold change ± SEM from 7–12 individual RNA samples from 2–3 independent experiments. (**f**) HA levels measured by ELISA assay in conditioned media. *, significantly increased in primary chemoresistant (*n* = 8) compared to chemosensitive (*n* = 10) serous ovarian cancer cells (*p* = 0.043, Mann Whitney U test). **, significantly increased in OV-90 CBPR conditioned media compared to parental cells (*p* = 0.0227, Mann Whitney U test).

**Figure 3 cancers-11-01187-f003:**
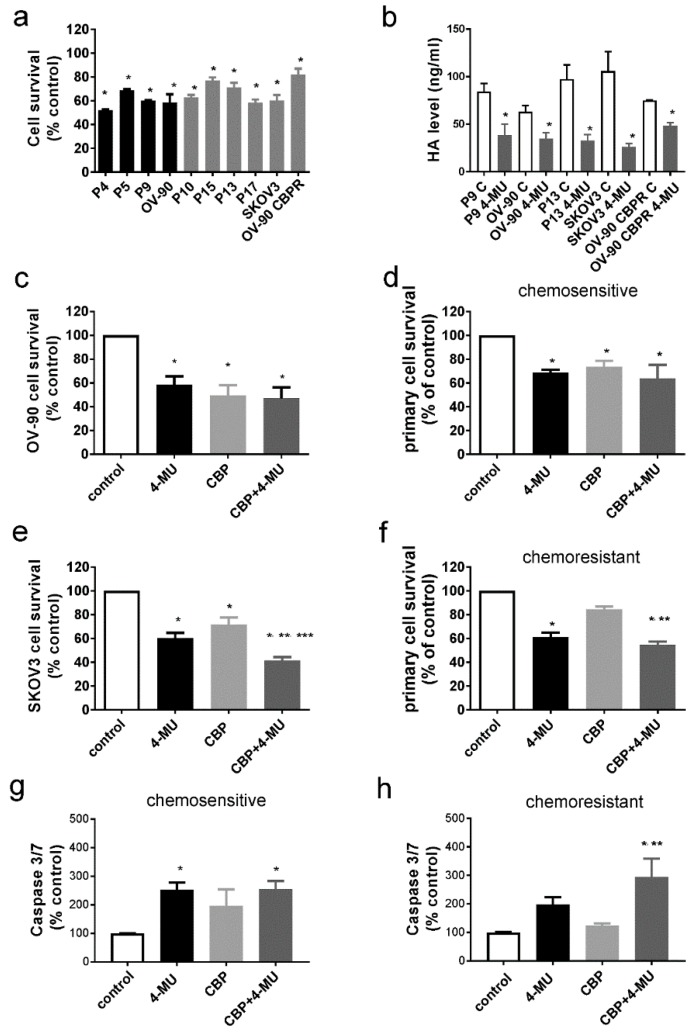
Effects of 4-methylubelliferone (4-MU) and carboplatin on ovarian cancer cell survival and HA production. (**a**) Ovarian cancer cell survival following 72 h of 4-MU (1 mM) treatment. The black and grey bars represent chemosensitive and chemoresistant cells, respectively. Data are expressed as percentage of control, mean ± SEM from 2–3 independent experiments performed in triplicate. *, significantly different from control media treatment (*p* < 0.05, independent *t* test). (**b**). Effect of 4-MU (1 mM) 72 h on HA production in conditioned media measured by HA ELISA. Data are expressed as ng/mL from 4–6 determinations. *, significantly different from control treatment (*p* < 0.05, independent *t* test). Effect of 4-MU on survival of OV-90 (**c**), chemosensitive primary cells (*n* = 2) (**d**), SKOV3 (**e**), and chemoresistant primary cells (*n* = 8), (**f**) assessed by MTT assay. Cells were treated with phosphate buffered saline (PBS), control, 4-MU (1 mM), carboplatin (CBP, 100 µM), and 4-MU (1 mM) + CBP (100 µM) for 72 h. Data are expressed as % of control from 2–5 independent experiments performed in quadruplicate. Effects of 4-MU (1 mM) and/or CBP (100 µM) treatment on apoptosis measured by caspase 3/7 cleavage in primary serous ovarian cancer cells. (**g**) chemosensitive (*n* = 3), and (**h**) chemoresistant (*n* = 5) cells. (**a**–**h**) *, significantly different from control, **, significantly different from CBP treatment, ***, significantly different from 4-MU treatment, (*p* < 0.05, one way ANOVA, Tukey’s multiple comparisons test).

**Figure 4 cancers-11-01187-f004:**
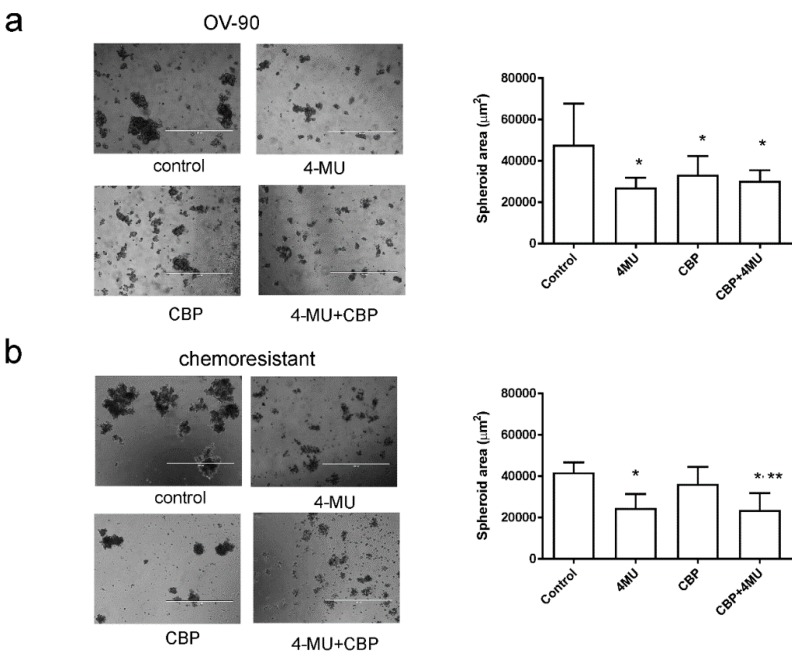
Effects of 4-methylubelliferone (4-MU) on spheroid formation. Representative images of spheroids formed by (**a**) OV-90 and (**b**) one of the chemoresistant primary ovarian cancer cells treated with control, 4-MU (1 mM), carboplatin (CBP, 100 µM), and 4-MU (1 mM) + CBP (100 µM) for 72 h. Data are expressed as median area of spheroids >200 µm in diameter (95% confidence interval, CI, *n* = 34–106) in five fields/treatment groups from 3–4 independent experiments. *, significantly different from control, **, significantly different from CBP treatment (*p* < 0.05, Kruskal–Wallis test, Dunn’s multiple comparison test). Scale bar = 1000 µm.

**Figure 5 cancers-11-01187-f005:**
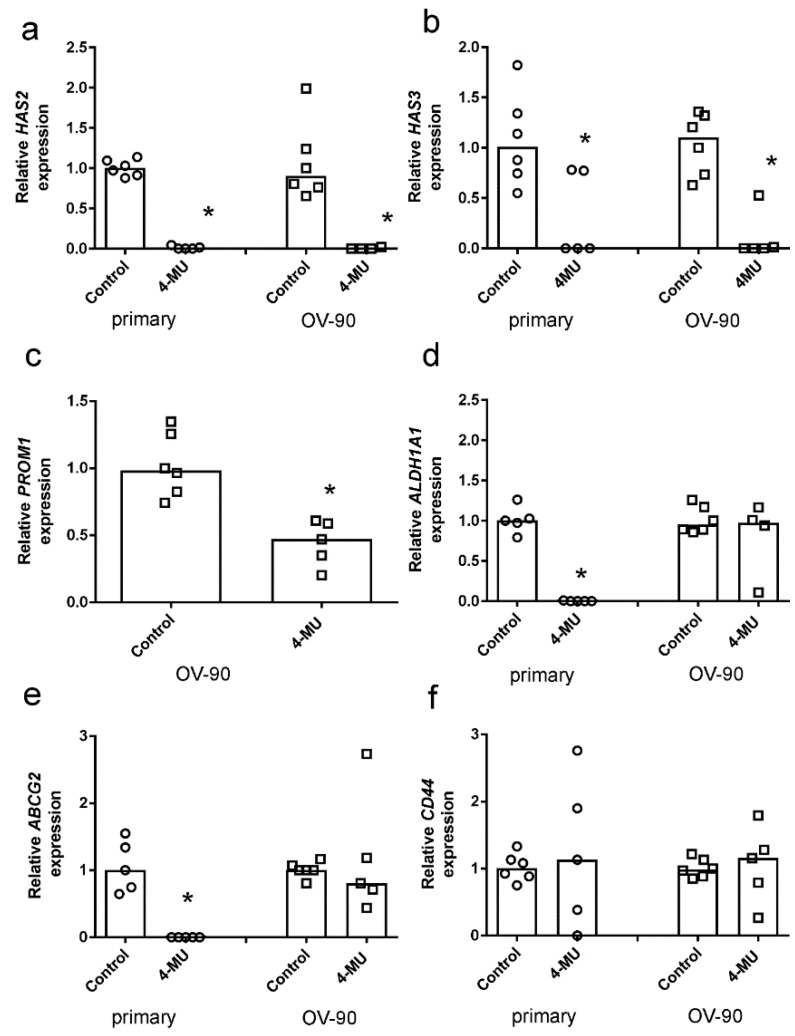
Effects of 4-methylubelliferone (4-MU) on stem cell marker expression in spheroids. Expression of stem cell markers in spheroids from a chemotherapy-resistant primary serous ovarian cancer and OV-90 cells treated with control media or 4-MU (1 mM) for 72 h. *HAS2* (**a**), *HAS3* (**b**), *PROM1* (**c**), *ALDH1A1* (**d**), *ABCG2* (**e**), and *CD44* (**f**). Data are expressed as the median from RNA samples (*n* = 5–6) from three independent experiments. *, significantly different from control, (*p* < 0.05, Mann Whitney U test).

**Figure 6 cancers-11-01187-f006:**
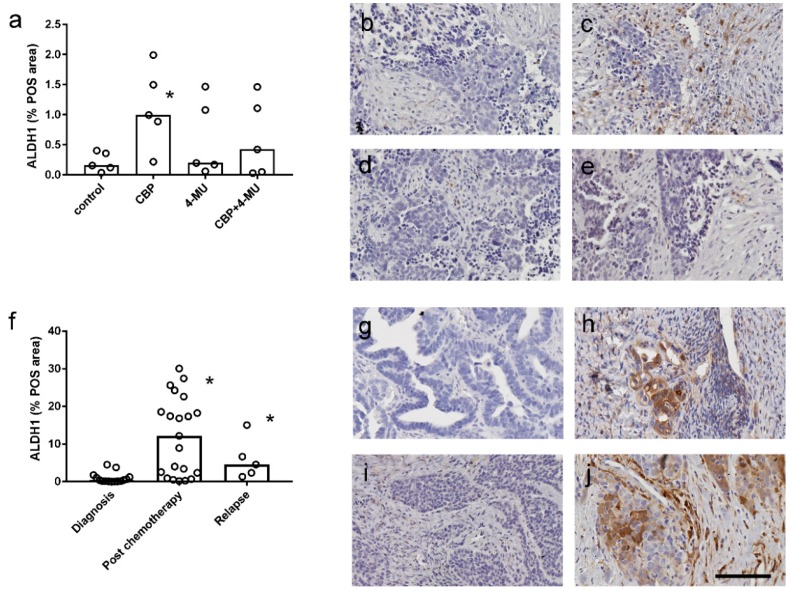
Effect of chemotherapy and 4-methylubelliferone (4-MU) on ALDH1 immunostaining in ovarian cancer tissues. (**a**) Quantitation of ALDH1 immunostaining in serous ovarian cancer explants tissues (*n* = 5) treated with control media, carboplatin alone (100 µM), 4-MU alone (1 mM), and CBP + 4-MU in combination. Data are expressed as percentage positive area (% POS) measured by video image analysis. *, Significantly different from control treatment (*p* = 0.0087, Friedman test, Dunn’s multiple comparison test). Representative images of ALDH1 immunostaining in the explant tissues treated with control media (**b**), CBP (**c**), 4-MU (**d**), and CBP+4-MU (**e**). (**f**) Quantitation of ALDH1 immunostaining in serous ovarian cancer tissues at diagnosis (*n* = 15), following neoadjuvant chemotherapy (*n* = 20) and at recurrence (*n* = 5). Data are expressed as % POS area. *, Significantly reduced from level at diagnosis (*p* = 0.0002, Kruskal–Wallis test, Dunn’s multiple comparison test). ALDH1 immunostaining in matched patient tissues at diagnosis (**g**) and following neoadjuvant chemotherapy (**h**). ALDH1 immunostaining in matched patient tissues at diagnosis (**i**) and at relapse with chemoresistant disease (**j**). All images are the same magnification. Scale bar = 100 µm.

**Figure 7 cancers-11-01187-f007:**
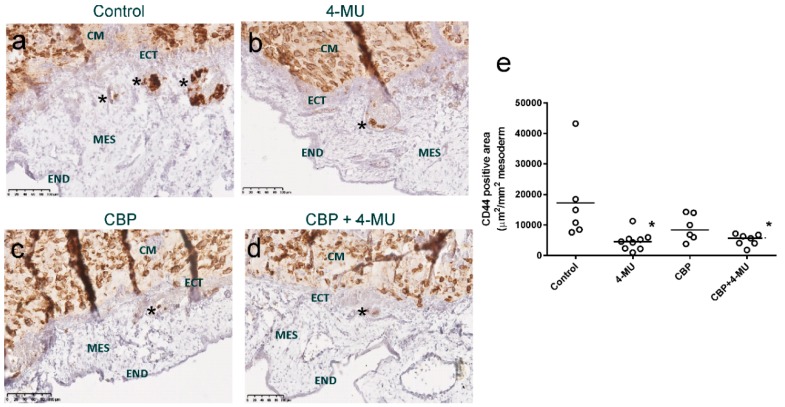
4-MU and carboplatin treatment inhibit in vivo invasion of chemoresistant primary cells. Representative images of the invasion of chemoresistant primary ovarian cancer cells shown by CD44 immunostaining. Treatment includes (**a**) control media, (**b**) 4-MU (1 mM), (**c**) carboplatin (CBP 100 µM, and (**d**) CBP 100µM + 4-MU (1 mM). Paraffin sections were immunostained with a CD44 mouse monoclonal antibody. Asterisks show examples of CD44 positive cancer cells that have invaded into the mesoderm (MES). (**e**) Quantitation of chorioallantoic membrane (CAM) invasion into the mesoderm. Data are of the CD44-positive area (µm^2^/mm^2^ of mesoderm) from 5–9 chick embryos implanted with chemoresistant primary cells from one patient. *, Significantly different from control (*p* = 0.0272, Kruskal–Wallis test, Dunn’s multiple comparison test). CM = cancer cells in matrigel implant, ECT = ectoderm, MES = mesoderm, END = endoderm. Scale bar = 100 µm.

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
