# Peer review of "4-Methylumbelliferone Inhibits Cancer Stem Cell Activation and Overcomes Chemoresistance in Ovarian Cancer"

_cancers, 2019, doi:10.3390/cancers11081187_

Round 1

Reviewer 1 Report

In this study, the authors investigated whether the hyaluronan (HA) synthesis inhibitor, 4-methylumbelliferone (4-MU) was effective in chemoresistant ovarian cancer cells. HA in patient’s serum was found to be correlated with chemoresistance, and 4-MU was found to enhance chemosensitivity and suppress invasion of chemoresistant ovarian cancer cells via the inhibition of HAS2 and HAS3 at the mRNA and serum HA levels. Interestingly, 4-MU suppresses the expression of cancer stem cell (CSC) markers and inhibits the ability of sphere formation, which is one of the features of CSC.

There is abundant data in literature supporting the role of HA in malignant cancers, but none were related to ovarian cancer. Thus, this study does provide additional data as it utilizes multiple ovarian cancer cell lines and patient-derived cells. In addition, CSC is regarded as a poor prognostic marker in cancer therapy, and this report suggested that it could be improved by 4-MU. Nevertheless, the manuscript could be strengthened in several regards. HA is functionally diverse due to changes in its molecular weight. For example, high-molecular-weight hyaluronan (HMWHA) has anti-inflammatory effects, while low-molecular-weight hyaluronan (LMWHA) is associated with angiogenesis. The recent studies on hyaluronan focused on its molecular weight. I do understand that not all of the below recommended changes are feasible in a manuscript revision. My recommendations are below.

Major comment:

1. In figures1 and 2f, are there any results available from dividing the hyaluronan in serum as HMW and LMW? If this is provided, the manuscript would be strengthened substantially.

2. As shown in figures 5a and 5b, 4-MU suppressed the mRNA expression of HAS2 and HAS3 in chemoresistant cells. HAS2 is reported to be involved in HMWHA synthesis and HAS3 is involved in LMWHA synthesis. Have you verified the effect of 4-MU on HA molecular weight?

3. HA is correlated with chemoresistance, but what is the causal relationship? I believe a detailed description should be provided in the discussion section. Similarly, an explanation on the relationship between CSC and HA could be valuable.

4. Does the administration of 4-MU elicit a dose-response curve? In this study, the 4-MU concentration mentioned is 1 mM. This concentration, in my opinion, is too high.

Minor comment:

1. In the material and methods section, under the spheroid assays, it should be stated whether the culture is serum free or not, and the status of growth factors such as EGF and bFGF etc could be elaborated.

2. I think 4-MU is poorly soluble. How did you use 4-MU?

Author Response

In figures1 and 2f, are there any results available from dividing the hyaluronan in serum as HMW and LMW? If this is provided, the manuscript would be strengthened substantially.

 We thank the reviewer for this suggestion. The HA ELISA we used in this study detects all forms of HA (15kDa->750kDa). We are very interested in determining HA molecular weight in serum and conditioned media from patients and ovarian cancer cells. This is an on-going investigation and methods for this are currently being set up in our laboratory.

As shown in figures 5a and 5b, 4-MU suppressed the mRNA expression of HAS2 and HAS3 in chemoresistant cells. HAS2 is reported to be involved in HMWHA synthesis and HAS3 is involved in LMWHA synthesis. Have you verified the effect of 4-MU on HA molecular weight?

 We are planning to determine effects of 4-MU on HA molecular weight in future studies.

HA is correlated with chemoresistance, but what is the causal relationship? I believe a detailed description should be provided in the discussion section. Similarly, an explanation on the relationship between CSC and HA could be valuable.

We have added a paragraph relating HA to chemoresistance in the revised manuscript (lines 307-318)

‘A potential mechanism whereby HA mediates chemoresistance is via expression of ABC transporter membrane proteins which decrease levels of chemotherapy drugs within cells [25]. Several studies have demonstrated that HA-CD44 interactions mediate chemotherapy resistance by regulating the expression and activity of ABC transporters which function as efflux pumps and interfere with the intracellular accumulation and retention of chemotherapy drugs [11,26]. We have previously demonstrated that HA can regulate the expression of multiple ABC transporters including ABCB3, ABCC1, ABCC2, and ABCC3 carboplatin treatment increased ABCC2 and HA in OVCAR-5 cells [13]. In breast cancer overexpression of HAS2 was reported to stimulate ABCB1/MDR1 expression through the PI3K pathway, increasing resistance to doxorubicin [27]. 500kDa HA was also found to stimulate MDR1 expression via CD44 in breast cancer (MCF-7 cells), inducing resistance to doxorubicin, paclitaxel and etoposide [11,28]. HA (molecular weight not specified) also promoted expression of ABCC2 in non-small cell lung cancer cells [29]‘.

 We have expanded the discussion on HA and CSC (lines 355-362)

‘The stemness and expansion of CSCs are thought to be highly influenced by changes in the microenvironment and recent studies have highlighted a key role for HA in regulating CSC populations [40,41]. Excessive HA production allows breast cancer cells to revert to a stem cell state via the up-regulation of genes involved in regulating epithelial-mesenchymal transition [41]. HA has also been shown to promote the formation of CSC populations in breast cancer [41] and glioblastoma cell lines [42]. Additionally, HA activates genes associated with stemness in embryogenesis and interacts with CSCs to enhance stemness and therapy resistance [43]. Shiina et al. have shown that molecular weight of HA was important in promoting and maintaining stemness of CSCs in the head and neck cancer cell line HSC-3 [44]. 200kDa HA significantly promoted expression of cancer stem cell genes and spheroid formation and cisplatin resistance in ALDHhigh CD44v3high HSC-3 cells compared to 5, 20 and 700kDa HA [44].’

Does the administration of 4-MU elicit a dose-response curve? In this study, the 4-MU concentration mentioned is 1 mM. This concentration, in my opinion, is too high.

A new supplementary Figure (Fig. S2) has been included in the revised manuscript to show effects of lower doses of 4-MU. Lower doses of 4-MU (0.1mM or 0.5mM) did not have any significant effect on the survival of OV-90, SKOV3 or the chemosensitive (P9) or chemoresistant (P13) primary serous cancer cells. Therefore, we used 1mM 4-MU for all subsequent experiments.

The following sentence has been added to the revised manuscript (lines 140-144)

‘We initially tested a range of 4-MU concentrations (0-1mM) and found that 1mM, but not lower concentrations of 4-MU (0.1mM, 0.5mM), could significantly inhibit the survival of OV-90, SKOV3 cells, chemosensitive (P9) and chemoresistant (P13) primary serous ovarian cancer cells (Fig S2). We used 1mM 4-MU for all subsequent experiments.’

Minor comment:

In the material and methods section, under the spheroid assays, it should be stated whether the culture is serum free or not, and the status of growth factors such as EGF and bFGF etc could be elaborated.

We used RPMI media with 10% FBS for the spheroid assays. No additional growth factors including EGF or bFGF were added to the media. We added this information to the methods section (lines 443-444).

I think 4-MU is poorly soluble. How did you use 4-MU?

We used the sodium salt of 4-methylumbelliferone (M1508, Sigma Aldrich) which is soluble in PBS/or water  as indicated in the specification sheet at 50mg/ml.

Reviewer 2 Report

The following points should be addressed:

  While 4MU has been consistently used to inhibit HA synthesis in many studies, there is some concern that 4MU may have other side effects besides inhibiting HA synthesis that could affect outcome. A short paragraph addressing this aspect of 4MU use and its specificity  would strengthen the MS.

  While HA is clearly affected by 4MU, molecules that interact with HA such as the hyaladherens may also be impacted. A brief recognition of this possibility would add balance to this MS.

  The discussion is heavily weighted on confirming that 4MU works on chemoresistance but there is little discussion as to why. Perhaps a brief paragraph describing some suspected mechanisms might elevate interest in this MS.

  The authors might address whether there is any value in specifically analyzing the " outliars"  since their results are quite variable.

Author Response

While 4MU has been consistently used to inhibit HA synthesis in many studies, there is some concern that 4MU may have other side effects besides inhibiting HA synthesis that could affect outcome. A short paragraph addressing this aspect of 4MU use and its specificity  would strengthen the MS.

We have recently conducted MTT assays with 4-MU ± exogenous HA using primary ovarian cancer cells. We found that exogenous HA (222kDa & 1110 kDa, 10µg/ml) treatment could not reverse the effects of 1mM 4-MU in the MTT assays (Fig S3). A study by Lompardia et al 2013 (ref 23) found that co-treatment with 500µM 4-MU and 30-fold higher levels of HA (300µg/ml) partially reverted the effects of 4-MU in leukemic K5652 and Kv563 cells. Treatment with lower concentration of 4-MU (100µM) and HA 300µg/ml completely restored baseline conditions in the cell lines. The authors concluded that effects of low concentrations of 4-MU could be restored by exogenous HA but higher doses of 4-MU might trigger anti-proliferative signals independent of HA. A study by Arai et al 2011 found that exogenous HA (200µg/ml) did not neutralize effects of 4-MU on osteosarcoma cells including formation of cell matrix, or cell proliferation (ref 31). The authors concluded from this finding that HA may have different biological activity if presented to cells as free HA or if it was cell-associated. It is likely that 4-MU has both HA dependent and HA independent effects on ovarian cancer cells and whether this can be blocked by exogenous HA may be dependent on the way HA is presented to cancer cells and its interactions with other extracellular matrix proteins.  

The additional sentence has been added to the results section (lines 148-149)

‘However, exogenous HA (222kDa & 1110 kDa, 10µg/ml) treatment could not reverse the effects of 1mM 4-MU in the MTT assays (Fig S3).’

This paragraph has been added to the discussion (lines 334-345)

‘Our findings that the addition of exogenous HA could not reverse the effects of 1mM 4-MU in the MTT assays supports this finding that 4-MU may also have anti-tumor activity that is not dependant on HA. The study by Lompardia et al 2013 [23] found that co-treatment with 500µM 4-MU and 30 fold higher levels of HA (300µg/ml) partially reverted the effects of 4MU in leukemic K5652 and Kv563 cells. Treatment with lower concentration of 4-MU (100µM) and HA 300µg/ml completely restored baseline conditions in the cell lines. The authors concluded that effects of low concentrations of 4-MU could be restored by exogenous HA and but higher doses of 4-MU may trigger anti-proliferative signals independent of HA. In another study Arai et al 2011 also found that exogenous HA (200µg/ml) could not neutralize effects of 1mM 4-MU on osteosarcoma cells including formation of cell matrix, or cell proliferation [31]. They concluded from this finding that HA may have different biological activity if presented to cells as exogenous free HA or as endogenous cell-associated HA.’

While HA is clearly affected by 4MU, molecules that interact with HA such as the hyaladherins may also be impacted. A brief recognition of this possibility would add balance to this MS.

We thank the reviewer for bringing this to our attention. We have not examined effects of 4-MU on hyaladherins like versican or other ECM proteins in our assays.  A study by Keller et al 2012 found that 4-MU reduced both versican and fibronectin in trabecular meshwork cells of the eye (ref 39) but the effects could not be reversed by the addition of exogenous HA to the culture medium. The authors suggested that since exogenous HA could not reverse the effect of 4-MU, only de novo synthesized HA altered versican and fibronectin levels.  It is likely that 4-MU affect the synthesis and organization of other ECM components to mediate its anti-proliferative effects on ovarian cancer cells.

The following sentences have been added to the discussion (lines 345-351)

‘Keller et al 2012 found that 4-MU reduced both versican and fibronectin in trabecular meshwork cells of the eye [39] but the effects could not be reversed by the addition of exogenous HA to the culture medium. They suggested that since exogenous HA could not reverse effects of 4-MU, only de novo synthesized HA altered versican and fibronectin levels. It is likely that 4-MU may affect the synthesis and organization of other ECM components to mediate its anti-proliferative effects on ovarian cancer cells.’

The discussion is heavily weighted on confirming that 4MU works on chemoresistance but there is little discussion as to why. Perhaps a brief paragraph describing some suspected mechanisms might elevate interest in this MS.

We have added a paragraph on potential mechanisms explaining the effect of HA on chemoresistance (lines 307-318) and expanded the discussion on HA and CSC (lines 355-362)

The authors might address whether there is any value in specifically analyzing the " outliers"  since their results are quite variable.

It is unclear which data the reviewer is referring to.

Round 2

Reviewer 1 Report

I think that revised manuscript has been significantly improved, and it is scientifically valuable.

Therefore, I recommend accept in present form.